# Effect of Functionalization of the Polycaprolactone Film Surface on the Mechanical and Biological Properties of the Film Itself

**DOI:** 10.3390/polym14214654

**Published:** 2022-11-01

**Authors:** Yuliya Nashchekina, Alina Chabina, Olga Moskalyuk, Irina Voronkina, Polina Evstigneeva, Gleb Vaganov, Alexey Nashchekin, Vladimir Yudin, Nataliya Mikhailova

**Affiliations:** 1Center of Cell Technologies, Institute of Cytology of the Russian Academy of Sciences, Tikhoretsky pr. 4, 194064 St. Petersburg, Russia; 2Laboratory of Polymer and Composite Materials «SmartTextiles», IRC–X-ray Coherent Optics, Immanuel Kant Baltic Federal University, 236029 Kaliningrad, Russia; 3Biochemistry Department, Institute of Experimental Medicine, Akad. Pavlov Street, 12, 197376 St. Petersburg, Russia; 4Institute of Macromolecular Compounds of Russian Academy of Sciences, V.O., Bol’shoy pr. 31, 199004 St. Petersburg, Russia; 5Laboratory «Characterization of Materials and Structures of Solid State Electronics», Ioffe Institute, Polytekhnicheskaya str., 26, 194021 St. Petersburg, Russia; 6Institute of Biomedical Systems and Biotechnology, Peter the Great St. Petersburg Polytechnic University, Polytechnicheskaya st., 29, 195251 St. Petersburg, Russia

**Keywords:** polycaprolactone films, arginine, mechanical properties, mesenchymal stem cells, extracellular matrix protein

## Abstract

The lack of suitable functional groups for cell adhesion on the surface of Polycaprolactone (PCL) is one of the main limitations in order to use PCL for biomedical applications. The aim of this research is to modify the PCL film surface using arginine, via an aminolysis reaction. In this regard, after PCL films formation by casting method, they were immersed in arginine solutions of various concentration at room temperature or then heated to 40 °C and in the presence of isopropanol or without it. To assess the structure of the modified surface, its wettability, and mechanical properties, methods of measuring the contact angle and the strip tensile test were used, and to compare the degree of attachment and the rate of cell proliferation, the method of fluorescent staining of cultured cells was used. The change in protein synthesis by cells on the modified surface was assessed using Western blotting. The results obtained show that the treatment of PCL films with an aqueous solution of arginine at room temperature for 1 day increases the hydrophilicity of the surface. Wherein surface modification led to a two-fold decrease of mechanical strength and flow stress, but elongation increase by about 30% for PCL films after modification in 0.5 M aqueous arginine solution at room temperature. Moreover, cell attachment and proliferation, as well as collagen synthesis, were significantly enhanced after arginine modification. The proposed simple and effective method for modifying PCL films with arginine significantly expands the possibilities for developing biocompatible scaffolds for tissue engineering.

## 1. Introduction

Poly(ε-caprolactone) (PCL), one of the biodegradable polyesters, has been widely used for tissue regenerating [1,2] to overcome the drawbacks of synthetic polyesters such as poor hydrophilicity and absent cellular recognition sites [3,4]. Since the surface of a biomaterial is in direct contact with the biological environment, foreign body reactions completely depend on surface characteristics of this biomaterial [5]. There are many techniques for surface modification of hydrophobic PCL scaffolds. The most common are physical coating, plasma etching, hydrolysis to physically and chemically immobilize bioactive compounds such as peptides, proteins and polysaccharides [6,7]. Among these methods the aminolysis is more convenient compared to monomer design and synthesis while brings minimum influence of bulk properties.

There are a large number of ester groups (-COO-) in PCL molecules. These groups can be hydrolysed to carboxylic acid under aminolysis [8]. The amino groups can be introduced onto the polyester surface by a reaction with diamine, providing that one amino group reacts with the -COO- group to form a covalent bond, -CONH-, while the other amino group is unreacted and free. Some advantages in tissue engineering can be expected by the steady introduction of these amino groups: (1) nontoxic to cells or tissues; (2) decreasing the surface hydrophobicity; (3) neutralizing the acid generated during the scaffold degradation and reducing the inflammation around the implanted scaffold; (4) providing active sites through which other biomolecules. 1,6-hexamethylenediamine and 1,2-ethylenediamine are used as a source of diamines [8,9].

Our research group is among the first to utilize L-arginine containing 2 terminal amino groups, to introduce primary amine groups to PCL [10]. Arginine is closely tied to several metabolic pathways involved in the synthesis of urea, polyamines, nitric oxide, agmatine, and creatine phosphate [11,12]. Our group and other researchers showed, that temperature and polarity of solvents noticeably influence the aminolysis process [13,14]. We have shown that the amount of arginine bound to the polymer surface increases at elevated temperatures. Moreover, we have shown that the presence of isopropyl alcohol increases the intensity of the aminolysis reaction. However, it should be noted that, in parallel with the aminolysis reaction, the hydrolysis reaction also takes place. As a result, ester bonds also dissolve, and carboxyl COO and hydroxyl OH groups form. Heating and isopropyl alcohol accelerate the hydrolysis reaction. As a result modification of polycaprolactone films with arginine increases the number of adhering cells and the number of focal contacts after 1 day of cultivation.

The question of degradation of films during their treatment with a modifying agent in an aqueous solution always remains open and important, since even partial degradation of PCL films leads to a change in their mechanical characteristics. Degradation of poly(α-hydroxyesters) and, in particular, PCL is stimulated by the specific action of water. Two mechanisms of PCL decomposition are considered in the literature. The first mechanism is characterized by the destruction of the surface of the polymer film, while the rate of hydrolytic splitting of macromolecules on the surface of the film is significantly higher than the rate of diffusion of water into the volume of the film. The second mechanism, on the contrary, is characterized by a sufficiently rapid penetration of water molecules into the volume of the film and, consequently, a uniform hydrolytic degradation over the entire volume of the PCL film [15]. PCL is characterized by hydrolytic degradation, as a result of which the size of polymer chains is reduced, carboxyl end groups are formed and water-soluble decomposition products such as 6-hydroxylcaproic acid are obtained [16]. During hydrolysis, degradation products can both escape into the surrounding water environment and diffuse into the polymer film and, thus, catalyze the process of volumetric degradation of the polymer film [17]. Mathematical models for predicting the rate of hydrolytic degradation of polyesters are presented in the scientific literature [18]. Previously, the results of both volumetric degradation of PCL films and PCL films with a modified surface were also published [19].

As shown above, in the process of the aminolysis reaction and under the action of isopropyl alcohol presented in the reaction medium, there is a rupture of ester bonds in the result of which the macromolecule is shortened. It can be expected that as a result of such treatment, not only the surface properties of polycaprolactone films, but also the mechanical properties change. The question of the further functioning of cells on modified films also remains unresolved. Adhesion and proliferation of cells on polymer substrates are certainly important, but they are not the only parameters for assessing the interaction of MSCs cells with the polymer surface. To form tissue in vitro after attachment, cells must themselves synthesize extracellular matrix proteins.

In this study, we clarify the following major concerns: (1) how do aminolysis reaction conditions, namely arginine concentration, temperature and polarity of solvent, affect on mechanical properties of PCL films; (2) its effect on the surface properties of PCL films, namely, hydrophilicity and the ability to sorb proteins; and (3) how the formed functional groups effect on the MSCs’ behaviour during long-term cultivation. Understanding these basic principles is essential to better use the arginine modification method to develop advanced biomaterials.

## 2. Materials and Methods

### 2.1. PCL Film Formation and Aminolysis

Polymers films were prepared by casting techniques [10,13]. PCL polymer powder (Mn = 80.000 g/mol; Sigma-Aldrich, St. Louis, MO, USA) was dissolved in chloroform (Reactiv, Saint-Petersburg, Russia).

### 2.2. Water Contact Angle

The static contact angles of water were measured at room temperature on a DSA30 contact angle measuring system from Kruss (Hamburg, Germany), using the sessile drop method. The essence of the measurement was as follows. A film sample was glued onto a glass slide. Then distilled water was applied with a special automatic dispenser on the test sample in an amount of 15 μL. Using the device’s camera, the droplet shape on the test sample is evaluated and the contact angle is calculated.

### 2.3. Evaluation of the Mechanical Properties of the Films

Mechanical tensile tests of the films at room temperature were carried out by using an INSTRON 1122 universal tensile testing machine. The test speed was 10 mm/min. For each film type, 5 samples having a base length of 30 mm were tested. The values of the tensile strength, yield point, Young’s modulus and deformation at break were calculated from the recorded stress–strain curves. Strain range for linear fit for Young’s modulus calculation was 0.025–0.25%.

### 2.4. Cultivation of MSCs

MSCs were isolated from the flat pelvis bones of rabbits using a modified version of previously described methods [10]. Bone marrow cells were cultivated in α-minimum essential medium (α-MEM; Lonza, St. Louis, MO, USA) supplemented with 10% fetal bovine serum (FBS; HyClone, St. Louis, MO, USA), 100 mg/mL streptomycin (Sigma-Aldrich, Steinheim, Germany), and 100 U/mL penicillin (Sigma-Aldrich, Steinheim, Germany).

### 2.5. Fluorescence Staining of MSCs

MSCs were fluorescence stained to study the effects of arginine modification on MSC adhesion and proliferation. In this experiment, unmodified PCL film was used as the negative control and pure glass was a positive control.

Cells were cultivated on the modified PCL films at 37 °C in a CO_2_ incubator for either 1 day or 3 day. Cell fixation and stain it was carried out according to the previously described method [13]. Cells were fixed with formaldehyde solution (Sigma-Aldrich, St. Louis, MO, USA) and Triton X-100 (Sigma-Aldrich, St. Louis, MO, USA). Rhodamine phalloidin (Thermo Fisher Scientific, Carlsbad, CA, USA) and DAPI (ab104139; Abcam, Cambridge, MA, USA) was used for cell stain. The actin cytoskeleton organization was then observed using a confocal microscope Olympus FV3000 (Olympus Corporation, Tokyo, Japan).

### 2.6. Cell Counts

To study the effects of the modified PCL films on cellular adhesion and proliferation, cells were cultured for a period of either 1 day, and 3 days, and then the cells were counted. For this purpose, five different pictures of fields on each matrix were taken at a wavelength of 365 nm (DAPI) using a fluorescence microscope Pascal (Carl Zeiss Jena GmbH, Jena, Germany). The ImageJ program was used to count the nuclei in each picture and cell counts were based on the number of colored cell nuclei in the pictures, and also to count the number of focal contacts [20].

### 2.7. The Determination of Synthesized Extracellular Matrix Protein

Protein expression at 3 days was analyzed by Western blotting. Sample preparation for analysis was performed as follows: an aliquot of conditioned medium after the cultivation was mixed with Laemmli buffer. For analysis of gel or scaffold material we take equal sample volumes adding the same buffer to said sample and processing further as other samples. After electrophoresis on 10% polyacrylamide gel transfer of proteins was carried out overnight in Towbin transfer buffer. The membrane was washed 3 × 5 min with TBS buffer (Tris-buffered saline, 25 mM Tris, 150 mM NaCl, pH 7.4) with 0.1% Tween-20, followed by blocking with BLOTTO solution (sc-2335, Santa Cruz Biotechnology, Dallas, TX, USA) for 30 min. After washing with TBS 3 × 5 min the membrane was incubated for 24 h at 8 °C with first antibodies then washed three times for 5 min in TBS buffer with 0.1% Tween-20 and incubated for 45 min at room temperature with secondary antibodies. Then membrane was washed 2 × 5 min in TBS solution with 0.1% Tween-20 followed by addition of BCIP/NBT substrate solution, (Sigma, V3804), and color development was observed for 2–15 min. Reaction was stopped; membrane was washed with water for another 20 min and scanned.

For ECM proteins identification the following antibodies were used: rabbit polyclonal antibodies to human plasma fibronectin, clone fn-15 (F7387, Sigma, St. Louis, MO, USA), mouse monoclonal antibody to human cellular fibronectin, clone fn-3E2 (F6140, Sigma, St. Louis, MO, USA), rabbit polyclonal antibody to human laminin (AB19012, Chemicon, UK), goat polyclonal antibodies to bone morphogenic protein (BMP) hBMP-2/4 (AF355, R&D, Minneapolis, MN, USA), rabbit polyclonal antibodies to collagen type I (AB745, Chemicon, UK) collagen type I, mouse monoclonal antibodies to collagen type III (C7805, Sigma, St. Louis, MO, USA) and mouse monoclonal antibodies to collagen type IV (MAB3326, Sigma, St. Louis, MO, USA), rabbit antibody to mouse IgG, conjugated with alkaline phosphatase (A3562, Sigma, St. Louis, MO, USA) and goat antibody to rabbit IgG, conjugated to alkaline phosphatase (A3687, Sigma, St. Louis, MO, USA) were used.

For quantitative analysis, the scanned images were processed using QuantiScan program, obtaining the value of image density in stained areas for corresponding proteins. The results were expressed in arbitrary units, corresponding to results of square of colored pixels multiplied on value of optical density of stained pixels.

### 2.8. Statistical Analysis

All experiments were performed in 3–5 replicates. A T-test was performed using Microsoft Excel software to analyze the statistically significant differences between specific samples. Samples were considered to be statistically important with the *p* < 0.05.

## 3. Results and Discussion

In our previously published works, the conditions of interaction of arginine with polycaprolactone film have been studied in detail [10,13]. It has been demonstrated that the amount of arginine, an increase in the modification temperature to 40 °C, as well as the presence of isopropyl alcohol affects the ability of arginine to bind to the polycaprolactone film. The largest amount of arginine was identified by the ninhydrin reaction on a film treated in 0.25 M arginine solution at a temperature of 40 °C for 1 h in the presence of isopropyl alcohol. Simultaneously with the aminolysis reaction, the hydrolysis of the PCL film surface also occurs. As a result of these reactions, functional hydroxyl and carboxyl functional groups are formed on the surface. The presence of these groups on the surface of the polymer film should increase the hydrophilicity of the surface. To assess the hydrophilicity of the surface, the water contact angle of wetting was measured.

### 3.1. Water Contact Angles

The water contact angles on the PCL films measured by a sessile drop technique are listed in Figure 1. The hydrophilicity of the PCL films was enhanced significantly after surface treatments by hydrolysis and aminolysis.

In Figure 1 it is seen the contact angle of films processed at room temperature depends slightly on the presence of arginine or alcohol in the solution. When the PCL films are incubated in pure water for 1 day, the contact angle decreases by almost 10 degrees to 58 degrees. The greater hydrophilicity of these films was attributed to the additional carboxylate (-COO-) and hydroxyl (-OH) groups on the PCL surface, which were initially resulted from the hydrolysis. It has been established, that alkaline hydrolysis increases surface hydrophilicity [21]. Under these conditions the influence of water molecules makes the largest contribution to the hydrophilicity of PCL films increase. Only the presence of a high concentration of arginine in an aqueous solution (0.5 M) reduces the wetting angle by another 2 degrees and reach angle as large as 56 degrees. Isopropyl alcohol also makes the same insignificant contribution to reducing the wetting angle. Thus, the presence of alcohol in the solution helps to reduce the wetting angle by 2–3 degrees.

Modification of PCL films by arginine solution when heated to 40 °C both arginine and isopropyl alcohol, make a significant contribution to the increase in hydrophilicity (Figure 1b). In contrast to treatment at room temperature for 24 h (Figure 1a), treatment upon heating after 1 h leads to a decrease in the contact angle by 5 degrees both in water and in the presence of isopropyl alcohol. In contrast to processing at room temperature, the aminolysis reaction, namely arginine, makes a significant contribution to the increase in the hydrophilicity of PCL films. Films treated with a solution with a maximum arginine content (0.5 M) have a maximum hydrophilicity. The contact angle of PCL films modified with a 0.5 M solution of arginine is 47 degrees. It should be noted that the conditions we developed for the modification of PCL films in the presence of arginine make it possible to obtain surfaces with a high degree of hydrophilicity. Moreover, the contact angle of PCL-modified arginine films is lower compared to similar films treated with other diamines. So H. Zhang, S. Hollister has been shown that PCL films treated with 1,6-hexanediamine have a contact angle of 66 degrees [22]. The modification of such RGD films with peptides allows one to obtain polymer wetting surfaces of which is 44 degrees. Yuan et al. also treated PCL films with a 1.6-hexanediamine solution and showed that this treatment reduces the contact angle to 67 degrees. But films with higher hydrophilicity can be obtained only after additional immobilization of collagen molecules on the amino groups. The contact angle of collagen-modified PCL films was 44 degrees [23].

The modification of films by isopropyl alcohol with heating does not significantly contribute to the contact angle value. Only a combination of isopropyl alcohol and a 0.25 M arginine solution will increase the hydrophilicity of modified PCL films corresponding to a contact angle of 57 degrees. It should be noted that the contact angle data for arginine-modified films are also lower compared to the contact angles of films treated with other diamines [23,24].

### 3.2. Mechanical Tests

The stress-strain curves of PCL films in their original form and after modification are shown in Figure 2. As can be seen, all curves have a similar character, consisting of two sections: the first section—the curve has a linear growth character, this is due to an increase in the stiffness of the polymer under tension, the second section—the deformation increases with constant stress, this is due to the fact that orientation processes occur in the polymer.

It should be noted that the modification leads to a decrease in the voltage value for all samples under the influence of all modifying factors, starting with water treatment at room temperature and ending with an arginine solution in an aqueous alcohol solution when heated to 40 °C. If in the control sample we observed voltage values within 9 MPa, then when exposed to an aqueous solution of arginine both at room temperature and when heated, the voltage values decrease by half for some samples. Moreover, both in the presence of water and alcohol, these values decrease significantly. We assume that such a significant decrease in voltage is due to a sufficiently active not only surface, but also volumetric degradation of the PCL film.

Comparison of the mechanical properties of the studied samples is presented in Figure 3. The initial PCL sample has the highest tensile strength σ_p_ = 9.5 MPa. Modification of PCL in distilled water at 40 °C allows maintaining the tensile strength of the sample almost at the initial level, σ_p_ = 8 MPa. Modification of PCL in distilled water at room temperature decreases the tensile strength of the sample in 3 times compered with initial PCL, σ_p_ = 3 MPa. Almost the same change in tensile strength is observed for PCL samples modified with a water-alcohol solution and a water-alcohol screen in the presence of 0.25 M arginine. When modifying these samples at a temperature of 40 °C, the σ_p_ is 5 MPa, when modified at room temperature, the tensile strength decreases by 4 and 3 MPa, respectively. As is true for all specimens described, modification at 25 °C results in a more significant reduction in tensile strength compared to modification at 40 °C. While for a sample modified with water and 0.5 M an increase in temperature leads to a decrease in the tensile strength of the sample by 2 times. σ_p_ modified 0.5 M arginine in water at room temperature is 6 MPa, and when modified at 40 °C σ_p_ = 3 MPa. As a result of the measurements, data were obtained indicating a decrease in the tensile strength for almost all films after modification. Indeed, it is known that the degradation of PCL is stimulated by the specific action of water and the phenomenon of diffusion and reactions that occur within the material. Two distinct processes are observed: (1) a surface degradation where the diffusion of water in the polymer volume is extremely slow compared to the hydrolytic cleavage reaction and (2) mass degradation where water is able to penetrate through the entire polymer so that random hydrolytic chain splits take place uniformly throughout the matrix [25]. The degradation of PCL is carried out by hydrolysis: the PCL chains are cleaved at the ester bonds, forming carboxyl terminal groups and thereby reducing the size of the molecular chains and yielding water-soluble degradation products such as 6-hydroxylcaproic acid [16,25,26]. According to the tensile strength data, the samples had the lowest values of this parameter after their processing at 25 °C during 8 h. Apparently, this is due to the duration of processing at room temperature, compared with 1 h of processing at 40 °C. With longer processing, in addition to the influence of temperature, arginine and isopropyl alcohol concentrations, degradation by-products also effect. When the by-products diffuse into the medium, the degradation, and, therefore, the decrease in the molecular weight throughout the sample are homogeneous, creating a balance between diffusion and hydrolysis reactions [27]. In the case where the by-products do not diffuse into the medium and remain trapped in the mass of the polymer, the presence of the carboxylic acids will catalyze hydrolysis [26].

The yield point of the initial and modified PCL in distilled water practically does not change and is 9 MPa, other types of modification lead to a 2-fold decrease in the yield point. We assume that in the presence of isopropyl alcohol, hydrolysis of PCL macromolecules occurs not only on the surface, but also in deeper layers of polymer films. The elongation at break of the original PCL is 100%.

The elongation at break of PCL samples modified at room temperature in distilled water and in a water-alcohol solution is at the same level. Increasing the modification temperature to 40 °C for these samples leads to a decrease in ε_p_ to 80%. The presence of arginine 0.25 M during modified in a water-alcohol solution leads to a decrease in the elasticity of PCL, and with an increase in temperature, this effect is more significant. The elongation at break of a sample modified at room temperature is 75%, at a temperature of 40 °C ε_p_ = 40%. Modification of PCL with 0.5 M arginine in distilled water solution at 40 °C makes it possible to maintain the elongation at break at the initial level, and even increase it by 30% at room temperature. The initial sample of PCL and modified in distilled water at a temperature of 40 °C have a Young’s modulus of 125 MPa. The use of other types of PCL modification leads to a decrease in the rigidity of PCL samples by an average of 30%. A similar dependence of the Young’s modulus on hydrolytic degradation was observed by other authors [15]. The authors of this study demonstrated a decrease in the Young’s modulus in the first period of hydrolytic degradation, but then the rate of decrease in the Young’s modulus decreases.

## 4. Adhesion and Proliferation of MSC on PCL Films

The cytotoxicity experiments confirmed the absence of in vitro cytotoxicity from the modified PCL films. The study of cell morphology showed that cells can still grow on modified surfaces, which is probably due to increased surface hydrophilicity and the presence of amino and hydroxyl groups. Moreover, it has been already demonstrated that arginine modification can modulate the MSC morphology, adhesion, and activity [10].

After 1 day of cultivation, the cells in all samples are well flattened and have an elongated fibroblast-like shape characteristic to MSCs (Figure 4). 3 days after seeding, the largest number of cells was on PCL films treated with arginine solution at heating (Figure 5). The cell proliferation after 3 days of cultivation on PCL films modified with an arginine solution in water is inversely proportional to the arginine concentration. In contrast to the results of proliferation, the number of cells after 24 h after seeding, the number of adherent cells on PCL films treated with arginine solution in water increased in proportion to an increase in the concentration of arginine solution. Thus, a high concentration of arginine on the PCL surface promotes cell adhesion to the modified surface in the first hours after seeding, but reduces proliferative activity during longer cultivation. The optimal amount of arginine on the film surface is 0.1 μg per 1 cm of surface.

## 5. Study of Protein Adsorption

We showed that different conditions for the aminolysis reaction and, therefore, different numbers of immobilized arginine amino groups affect the adhesion, proliferation, organization of the actin cytoskeleton of focal MSCs. How the surface modification of PCL films influence on protein adhesion? Before cells adhere on PCL surfaces, proteins in the culture medium and serum adsorb on the surfaces rapidly, and then these adsorbed proteins provide adhering sites for cells [28,29,30,31].

Fibronectin is present in a soluble form in plasma or sera of all vertebrate species and in growth media of cells cultured in vitro [32]. It is found in an insoluble form on the surface of normal cells and in the surrounding extracellular matrix. In order to characterize fibronectin adsorption on pristine and treated PCL surfaces in the present study, we applied Western blotting assay to quantify the amounts of adsorbed fibronectin. Figure 6 clearly indicates more fibronectin adsorption on the treated PCL films than on pristine PCL surface. It should be noted that the films after the reaction of both hydrolysis and aminolysis at room temperature well absorb fibronectin from the culture medium. The greatest amount of fibronectin is adsorbed onto the film, which, according to the results of the ninhydrin reaction, has the largest number of amino groups. On PCL films treated with a solution of arginine when heated, the largest amount of arginine is also adsorbed on films having the largest number of amino groups, namely films processed in a solution of arginine in the presence of isopropanol. It should be noted that the films after the reaction of both hydrolysis and aminolysis at room temperature well absorb fibronectin from the culture medium. The greatest amount of fibronectin is adsorbed onto the film, which, according to the results of the ninhydrin reaction, has the largest number of amino groups.

On PCL films treated with a solution of arginine when heated, the largest amount of arginine is also adsorbed on films having the largest number of amino groups, namely films processed in a solution of arginine in the presence of isopropanol. The figure shows that not only the aminolysis reaction positively affects the sorption of fibronectin, but also the hydrolysis reaction, as a result of which free carboxyl groups (COO-) are formed.

Our results are consistent with those in the pertinent literature: for instance, Arima and Iwata observed more proteins adsorbed and more cells adhered on -COOH and -NH_2_ surfaces than on -CH_3_ and -OH surfaces. [33]. Cao and coauthors confirmed less proteins adsorbed from the cell culture media on the neutral -CH_3_ and -OH surfaces than on the charged -COOH and -NH_2_ surfaces [34]. As most proteins are charged in the cell culture environment, surface charge of polymer surface could influence the protein adsorption very significantly. More proteins adsorbed on charged polymer surfaces is beneficial for the subsequent cell adhesion and spreading, which explains the larger spreading area of actin filaments and more adherent cells on the modified PCL surfaces in our experiments.

Figure 7 shows fibronectin, collagen I and collagen IV deposition on modified PCL surfaces. The effect of arginine modification of PCL films on fibronectin, collagen I and collagen IV deposition were studied by Western blotting assay.

Fibronectin is one of the major ECM proteins and plays an important role in the adhesion process of mammalian cells to polymer surface through specific interactions with the receptors of cellular membrane such as RGD peptides [33]. Therefore, it is important to investigate whether PCL surface with immobilized arginine could promote MSCs to synthesize fibronectin and facilitate the proliferation of the cells on the polymer surfaces. On day 3 after cell seeding, Western blotting assay show the maximum amount of deposited cellular fibronectin on PCL films hydrolyzed in water at room temperature and treated in 0.1 M arginine solution at water and at room temperature too (Figure 7). MSCs were seeded on PCL films treated at 40 °C demonstrated the maximum amount of deposited cellular fibronectin on polymer films modified by 0.1 M and 0.25 M arginine solutions in isopropanol. Films modification by arginine solutions such room temperature (25 °C) as 40 °C promote cellular fibronectin deposition compare to cells seeded on pristine PCL films or glass (Figure 7a).

Surface chemistry is known to affect the adhesion, proliferation, and migration of a variety of cells [35,36,37,38]. Wherein its effect on the deposition of both collagen I and IV has not been explored. Surface chemistry is also known to affect protein adsorption [39,40], while collagen deposition by cells is an active, not a passive process [41]. These data suggest that PCL surface, treated by arginine solutions in isopropanol, either encourages formation adhesion of MSCs focal contacts or more favorable for collagen I deposition in early stage of growth. Despite the fact that arginine modification improves the synthesis of collagen I by MSCs, it does not affect the deposition of collagen IV. In some cases, modification with arginine even leads to a decrease in the synthesis of collagen IV on the treated films compared to pristine PCL (Figure 7c).

## 6. Conclusions

Modification of the surface of PCL films was carried out by the reaction of aminolysis. The natural amino acid arginine was used as a source of amino groups. Varying the reaction conditions, namely the concentration of arginine, the reaction temperature and the presence of isopropyl alcohol, makes it possible to control the reaction of aminolysis and, consequently, the mechanical and biological properties of modified PCL films. Previously published works have proved the possibility and effectiveness of the proposed method of modification of hydrophobic PCL films. In this study, it was possible not only to expand the understanding of the influence of modification conditions on the properties of cultured cells, but also to study the properties of modified films in more detail. Thus, based on measurements of the mechanical characteristics of films, namely, a decrease in the strength characteristics of modified films at the initial moment of deformation loads, we made the assumption that both surface and bulk degradation of PCL films occurs during such modification. Moreover, the films are already subjected to the degradation process after treatment in clean water for 1 day. A noticeable decrease in yield strength (almost twice) compared to the original unmodified PCL sample was observed in PCL films treated with arginine solution. One of the reasons for such a sharp decrease in yield strength may be the presence of arginine both on the surface and inside the PCL film. As mentioned earlier, the decrease in stress in the samples after modification is a consequence of the volumetric degradation of PCL films, which contributes not only to the surface reaction of aminolysis, but also to the active absorption of arginine into the sample volume. And the presence of arginine molecules in the volume of PCL films leads to difficulty in their movement under stress, and, consequently, to a decrease in the yield strength, which we observe in Figure 2 and Figure 3. A decrease in the wetting angle in all modified films indicates the active formation of hydroxyl and carboxyl groups on the surface of PCL films. Another result indicating the volumetric degradation of PCL films in the process of such modification is their sufficiently high sorption capacity relative to fibronectin. Moreover, a significant amount of sorbed fibronectin is observed on samples modified at room temperature for 1 day. We assume that such results are associated with deeper volumetric degradation of films during 1 day at room temperature, compared with degradation over a shorter time interval, even when heated to 40 °C. Obviously, this result is related to the autocatalysis described earlier. According to the obtained results, 1 day after seeding, the largest number of cells were observed on PCL films treated with arginine in an aqueous alcohol solution. However, after 3 days of cultivation, the largest number of cells were on films treated in an arginine in an aqueous solution without alcohol. Thus, despite the decrease in a number of mechanical properties of modified films, the ability to adhesion, cell proliferation on such films, as well as the synthesis of extracellular matrix proteins significantly increases. Therefore, such films can be further used for cell cultivation and transplantation.

## Figures and Tables

**Figure 1 polymers-14-04654-f001:**
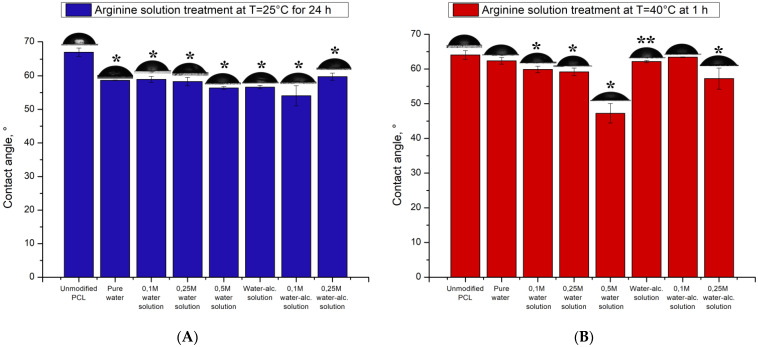
Water contact angle measurements for PCL surface before and after the hydrolysis and aminolysis treatment. (**A**)—Arginine solution treatment at T = 25 °C; (**B**)—Arginine solution treatment at T = 40 °C.

**Figure 2 polymers-14-04654-f002:**
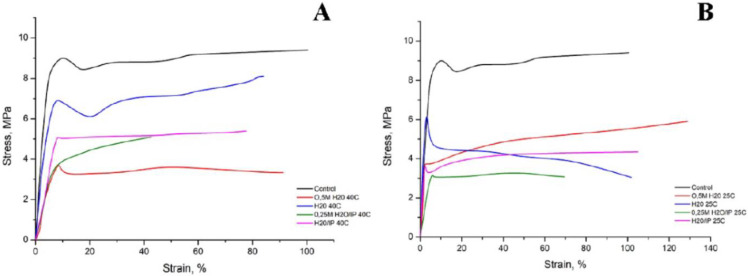
Stress-strain curves of PCL films before and after the hydrolysis and aminolysis treatment. (**A**)—40 °C и (**B**)—25 °C.

**Figure 3 polymers-14-04654-f003:**
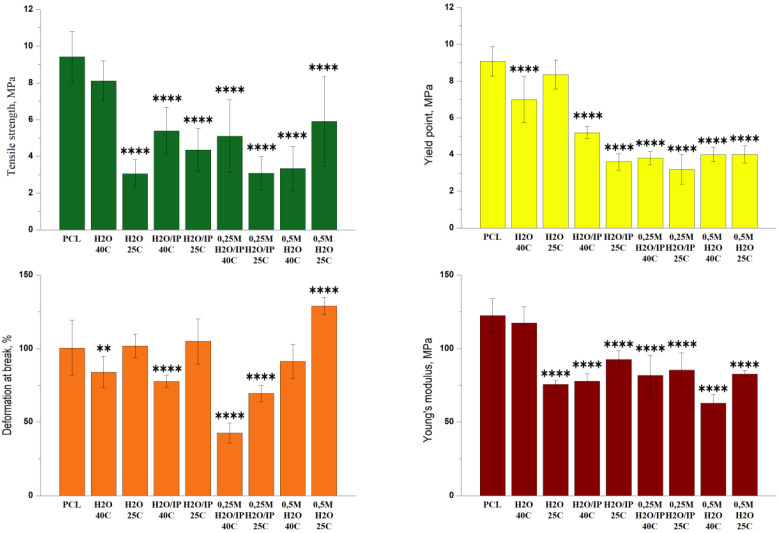
Mechanical properties for PCL films before and after the hydrolysis and aminolysis treatment. Significant differences are marked with ** (*p* < 0.01) and **** (*p* < 0.001).

**Figure 4 polymers-14-04654-f004:**
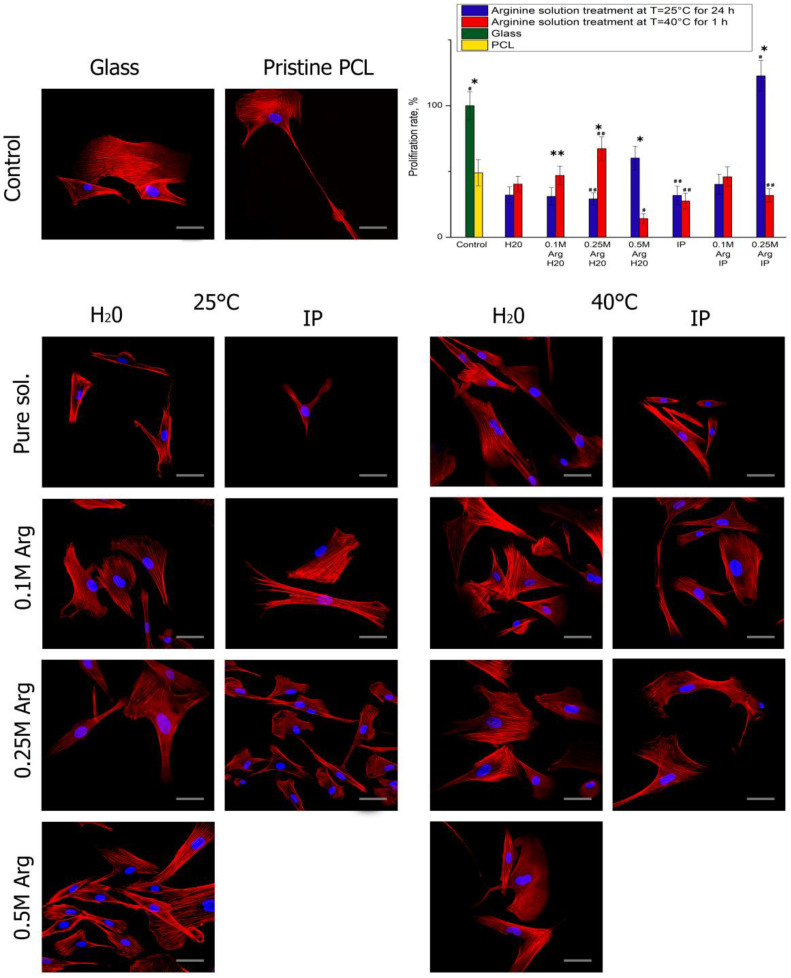
Fluorescence micrographs of cells (MSCs) stained after 1 day cultivation on PCL surfaces treated by hydrolysis and aminolysis reactions: actins (red) and nuclei (blue). Scale bar 50 µm. The diagram shows the proliferation rate of cell adhesion on surfaces with different functional groups after 1 day cultivation (n = 4). Significant differences are marked with * (*p* < 0.05), ** (*p* < 0.01) for the same concentration data, #—*p* < 0.05, ##—*p* < 0.01 compared with the unmodified PCL.

**Figure 5 polymers-14-04654-f005:**
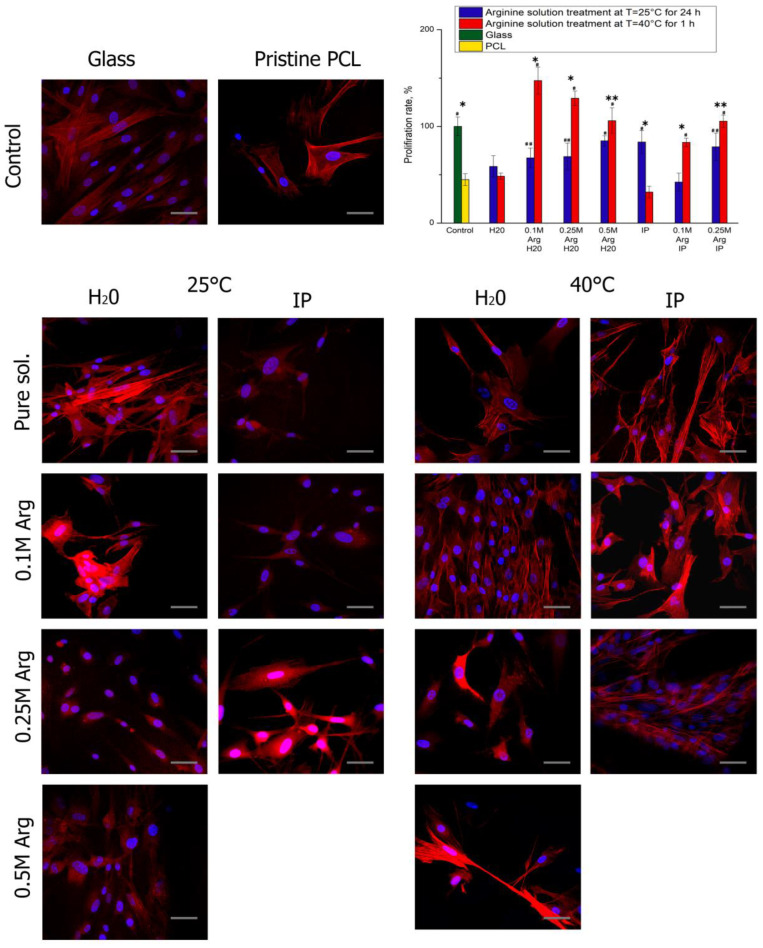
Fluorescence micrographs of cells (MSCs) after 3 days cultivation on PCL surfaces treated by hydrolysis and aminolysis reactions: actins (red) and nuclei (blue). Scale bar 50 µm. The diagram shows the proliferation rate of cell adhesion on surfaces with different functional groups after 3 days cultivation (n = 4). Significant differences are marked with * (*p* < 0.05), ** (*p* < 0.01) for the same concentration data, #—*p* < 0.05, ##—*p* < 0.01 compared with the unmodified PCL.

**Figure 6 polymers-14-04654-f006:**
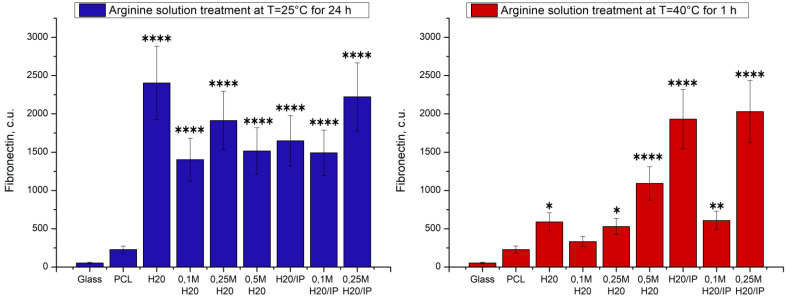
The diagram shows the fibronectin adsorption level. Significant differences are marked with * (*p* < 0.05), ** (*p* < 0.01) and **** (*p* < 0.001).

**Figure 7 polymers-14-04654-f007:**
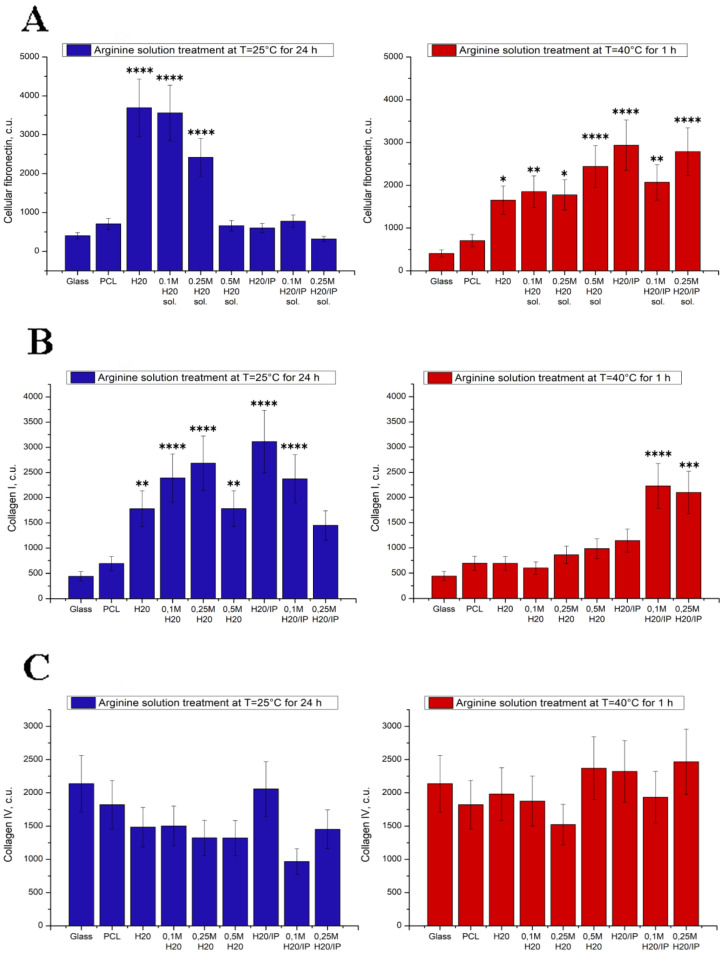
The diagrams show Fibronectin (**A**), collagen I (**B**) and collagen IV (**C**) deposition level by MSCs on modified PCL surfaces. Significant differences are marked with * (*p* < 0.05), ** (*p* < 0.01), *** (*p* < 0.001) and **** (*p* < 0.0001).

## Data Availability

Data available upon request.

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
