# Peer review of "Effect of Functionalization of the Polycaprolactone Film Surface on the Mechanical and Biological Properties of the Film Itself"

_polymers, 2022, doi:10.3390/polym14214654_

Round 1

Reviewer 1 Report

The manuscript reported the surface modification and characterization of PCL films, which discussed the parameters/component that optimized the mechanical property of this important polymer material. The results are well presented and shows some interesting insight, while some additions need to be made:
*/ The abstract and conclusion are conceived only in general terms. I recommend working out the conclusion in more detail - summarizing specific results; evaluate which of the surface treatment procedures was more suitable?

*/ For a complete evaluation of the surface treatment, it is necessary to supplement the publication with one of the surface/chemical analyses. I recommend supplementing the statement with FTIR analysis or SEM scans of surfaces.

*/ I lack a more detailed description of the behavior of the PCL material during the tensile test. It would be nice to have the strain rate here. The Mullins effect may play important roles depending on the applied strain rate.

Author Response

Response to Reviewer 1 Comments

Reviewer 1

The manuscript reported the surface modification and characterization of PCL films, which discussed the parameters/component that optimized the mechanical property of this important polymer material. The results are well presented and shows some interesting insight, while some additions need to be made:

Dear Reviewer 1, we are very grateful to you for manuscript review and valuable comments. We tried to take into account all your comments and recommendations and noticeably improve the manuscript. Thanks a lot.

*/ The abstract and conclusion are conceived only in general terms. I recommend working out the conclusion in more detail - summarizing specific results; evaluate which of the surface treatment procedures was more suitable?

Dear Reviewer 1, we have revised the conclusion in the manuscript and summarize the results in more detail. All corrections in the text are marked in yellow. According to the obtained results, 1 day after seeding, the largest number of cells were observed on PCL films treated with arginine in an aqueous alcohol solution. However, after 3 days of cultivation, the largest number of cells were on films treated in an arginine in an aqueous solution without alcohol. The presence of isopropyl alcohol also affected the sorption of fibronectin on modified PCL films. It was demonstrated that cells synthesize type I collagen to a greater extent on modified films compared to cells cultured on intact PCL films.

*/ For a complete evaluation of the surface treatment, it is necessary to supplement the publication with one of the surface/chemical analyses. I recommend supplementing the statement with FTIR analysis or SEM scans of surfaces.

We understand the importance of the results of FTIR and SEM analysis. We have already published these studies in our previous article (Nashchekina, Y.; Chabina, A.; Nashchekin, A.; Mikhailova, N. Different Conditions for the Modification of Polycaprolactone Films with L-Arginine. Int. J. Mol. Sci. 2020, 21, 6989, doi:10.3390/ijms21196989). Presented work is a continuation of the work on the modification and analysis of PCL films modified with arginine (Main results are mentioned below).

Figure. Fourier transform infrared spectra of the various films: (A) unmodified poly(“-caprolactone), (B) PCL treated with the 0.25 M arginine solution in water for 24 h at T = 25 °C, (C) PCL treated with the 0.25 M arginine solution in isopropanol for 24 h at T = 25 °C, (D) PCL treated with the 0.25 M arginine solution in water for 1 h at T = 40 °C, (E) PCL treated with the 0.25 M arginine solution in isopropanol solution for 1 h at T = 40 °C. The two additional peaks in the range from 1550 to 1650 cm−1 in aminolyzed PCL films indicated the introduction of amine groups onto the PCL substrates.

Figure. Scanning electron microscopy (SEM) images of PCL films; unmodified PCL film; film treated with 0.1 M arginine solution for 1 h at T = 40 °C; film treated with 0.25 M arginine solution for 1 h at T = 40 °C; film treated with 0.5 M arginine solution for 1 h at T = 40 °C; film treated with 0.1 M arginine solution for 24 h at T = 25 °C; film treated with 0.25 M arginine solution for 24 h at T = 25 °C; film treated with the 0.5 M arginine solution for 24 h at T = 25 °C. Scale bar 100 µm.

*/ I lack a more detailed description of the behavior of the PCL material during the tensile test. It would be nice to have the strain rate here. The Mullins effect may play important roles depending on the applied strain rate.

The section of mechanical tests has been deeply redesigned. We marked the corrected parts in yellow.

Reviewer 2 Report

Dear authors

The document has good novelty and could be useful for researchers. But results such as contact angles and mechanical properties are shown with charts and pictures related to the data are not included in the paper. This is the main scientific problem and the corresponding pictures should be included in the paper. The significant similarity as well as grammatical and spelling mistakes are the main problem related to paper writing. If your final decision is to publish the paper, it needs major revision and should be corrected significantly. Sincerely

Author Response

Response to Reviewer 2 Comments

Reviewer 2

The document has good novelty and could be useful for researchers. But results such as contact angles and mechanical properties are shown with charts and pictures related to the data are not included in the paper. This is the main scientific problem and the corresponding pictures should be included in the paper. The significant similarity as well as grammatical and spelling mistakes are the main problem related to paper writing. If your final decision is to publish the paper, it needs major revision and should be corrected significantly. Sincerely

Dear Reviewer 2, we are very grateful to you for manuscript review and valuable comments. We tried to take into account all your recommendations, expanded and revised the results obtained, which do allowed us to significantly improve the manuscript. Thanks a lot.

In the processed version, new results have been added to determine the wetting angle, as well as the section of mechanical tests has been deeply redesigned. We marked the corrected parts in yellow.

Author Response

Response to Reviewer 3 Comments

Reviewer 3

Dear Reviewer 2, we are very grateful to you for manuscript review and valuable comments. We tried to take into account all your recommendations, expanded and revised the results obtained, which do allowed us to significantly improve the manuscript. All corrections in the text are marked in yellow. Thanks a lot.

-plagiarism rate is 52%. This rate is unacceptable.

The information about the high percentage of plagiarism was unexpected for us. We wrote the entire text of the article ourselves, with the exception of standard methods. We have rewritten the methods in the new version.

-The introduction should be rewritten, the literature should belong to the last 5 years, and the results should be compared with related work.

Indeed, PCL has been investigated for a long time. We have rewritten the introduction and added references to the literature for the last 5 years.

-Generally used literature should include studies from more recent years rather than previous years.

We have tried to take this remark into account in the text.

-There are some errors in the spelling of units. for example liter should be shown as 'L' not ''l'' or cm2 should be shown as upper index

We have tried to take into account all the recommended comments in the text.

-All content should be edited in English, there are very obvious grammatical errors;

.And only....??????

.But......??????

It contains many errors like this

We have tried to take into account all the recommended comments in the text.

-The conclusion part is not explained in sufficient detail, the results of the analysis should be included and the discussion should be summarized.

We have revised the conclusion in the manuscript and summarize the results in more detail.

-How were the arginine ratios determined? What is the correlation between the ratios? Also, possible consequences should be discussed.

We have already published these studies in our previous article (Nashchekina, Y.; Chabina, A.; Nashchekin, A.; Mikhailova, N. Different Conditions for the Modification of Polycaprolactone Films with L-Arginine. Int. J. Mol. Sci. 2020, 21, 6989, doi:10.3390/ijms21196989). Presented work is a continuation of the work on the modification and analysis of PCL films modified with arginine (Main results are mentioned below).

Figure. The dependence of the arginine amount on the modified film per 1 cm2 on the solution concentration (water (H2O) or isopropanol (IP)) at T = 40 °C and T = 25 °C. (n = 5: *—p < 0.02 for the same concentration data, **—p < 0.05; #—p < 0.02 compared with lower concentration, but the same temperature).

In the revised version of the article, we tried to take into account these comments.

The influence of different arginine concentrations and solvent consisted of water and isopropanol (3:1) on the amount of immobilized NH2 groups were also investigated. The results demonstrated that the –NH2 group density on the PCL films increased along with the growth of arginine concentration in water or in solvent from isopropanol and water. The presence of isopropanol in the solvent, as well as heating the reaction solution, allows the maximum amount of amino groups to be immobilized on the surface of the PCL film. At room temperature, the number of immobilized amino groups on films treated in a 0.5 M aqueous solution of arginine is equal to the number of amino groups immobilized on films treated with a 0.25 M solution of arginine in the presence of isopropyl alcohol. However, upon heating, the number of amino groups immobilized on films treated in a 0.25 M solution of arginine in the presence of isopropanol is two times higher compared to films treated in a 0.5 M aqueous solution.

Round 2

Reviewer 1 Report

Dear authors.

I agree to the processing of the publication. However, I recommend that you correct typos and format the text correctly, for examples:

Figure 2./ Stress-strain curves of PCL films were not added;

line 284/ error 40C ...40°C.

Author Response

Dear Reviewer, we are very grateful to you for manuscript re-reviewing. Apparently an error occurred and Figure 2 was accidentally deleted. We have tried to make all the corrections in the new version of the article.

Reviewer 2 Report

Dear authors

The manuscript has high similarity to other sources.

Also, it has numerous spelling an grammatical errors.

The mechanical test graphs are not included in the paper.

Consequently, the mentioned points represent that the previous revision is not considered by authors completely. So, the paper must be rejected.

Sincerely

Author Response

Dear Reviewer, we have tried to take into account all the comments and correct the manuscript. All changes in the text were highlighted in yellow. Data on mechanical tests have also been added to the text.

Author Response

Dear Reviewer, we are very grateful to you for manuscript re-reviewing. We have tried to make all the corrections in the new version of the article.
